# Backhand-Approach-Based American Sign Language Words Recognition Using Spatial-Temporal Body Parts and Hand Relationship Patterns

**DOI:** 10.3390/s22124554

**Published:** 2022-06-16

**Authors:** Ponlawat Chophuk, Kosin Chamnongthai, Krisana Chinnasarn

**Affiliations:** 1Department of Electronic and Telecommunication Engineering, Faculty of Engineering, King Mongkut’s University of Technology Thonburi, Bangkok 10140, Thailand; ponlawat.baw@mail.kmutt.ac.th; 2Faculty of Informatics, Burapha University, Chon Buri 20131, Thailand; krisana@informatics.buu.ac.th

**Keywords:** American sign language words, computer vision, bidirectional long short-term memory (BiLSTM), video processing, backhand approach, portable system, leap motion sensor, SRM sign group, the spatial–temporal body parts and hand relationship patterns (ST-BHR patterns)

## Abstract

Most of the existing methods focus mainly on the extraction of shape-based, rotation-based, and motion-based features, usually neglecting the relationship between hands and body parts, which can provide significant information to address the problem of similar sign words based on the backhand approach. Therefore, this paper proposes four feature-based models. The spatial–temporal body parts and hand relationship patterns are the main feature. The second model consists of the spatial–temporal finger joint angle patterns. The third model consists of the spatial–temporal 3D hand motion trajectory patterns. The fourth model consists of the spatial–temporal double-hand relationship patterns. Then, a two-layer bidirectional long short-term memory method is used to deal with time-independent data as a classifier. The performance of the method was evaluated and compared with the existing works using 26 ASL letters, with an accuracy and F1-score of 97.34% and 97.36%, respectively. The method was further evaluated using 40 double-hand ASL words and achieved an accuracy and F1-score of 98.52% and 98.54%, respectively. The results demonstrated that the proposed method outperformed the existing works under consideration. However, in the analysis of 72 new ASL words, including single- and double-hand words from 10 participants, the accuracy and F1-score were approximately 96.99% and 97.00%, respectively.

## 1. Introduction

Sign language is a medium of communication for hearing-impaired people, a group which includes 466 million people worldwide [1], and it is expressed using the fingers and hands. American Sign Language is the most populous among its peers, and thus consists of over ten thousand word gestures [2]. Moreover, 65% of ASL gestures represent sign words during a full conversation [3]. In addition, communication between hearing people and hearing-impaired people is very important because they must work together; however, hearing people do not know much about sign language, so there exists a communication gap and this gap in communication occurs very often in society. For instance, in the case of medical diagnosis where the patient is hard-of-hearing, the information received from the patient can be inaccurate, which can affect critical healthcare decisions. Poor communication often has catastrophic effects. According to a World Health Organization report, the simple failure to follow doctor’s orders results in 125,000 deaths in the U.S. each year [4]. Moreover, unemployment statistics for persons with deafness, caused by a lack of communication, have reached 3.8% in the U.S. [5]. Therefore, an automatic sign language interpretation system is necessary to bridge a communication gap. Automatic sign language interpretation systems need to be portable and mobile; thus, a backhand approach using a portable sensing device was proposed in [6], with a sensor mounted on the user’s chest to make it portable and user-friendly.

Some authors, such as [7], have developed automatic sign language interpretation systems based on the relationship between hand positions and body part features to recognize dynamic word gestures. Ma et al. [7] used WiFi signals to detect the fingers, hands, arms, and head. Although this method showed good recognition performance, its use is restricted by the need for high-speed internet access and a good line of sight. However, dynamic sign language recognition using analysis of the hand and body relationship has demonstrated superb results, for example, in the visual recognition of traffic police gestures [8], bipolar disorder recognition [9] using upper body postures, and emotional stress state recognition [10]. However, using the relationship between hand positions and body part features may not be extendable to sign language used by the hearing-impaired group. This is because of the semantic nature of their sign language. In addition, these features are used with a forehand view, which leads to misclassification; therefore, it is difficult to apply this method with a backhand view. The authors in [2,3,6,11,12] extracted hand features using a backhand view approach, which led to outstanding performance; however, these studies may fail to recognize words with a similar shape, rotation, and movement (SRM words). To address this problem with the existing works of [2,3,6,11,12], in this paper we propose the spatial–temporal body parts and hand relationship patterns (ST-BHR patterns) as the main feature of analysis for 72 isolated signed words in the SRM group [13] based on the backhand approach. In this method, both single and double hands were applied, and their information was obtained using the 3D distance-based Cartesian product obtained from a 3D depth camera. In addition, three additional features, spatial–temporal finger joint angle patterns, spatial–temporal double-hand relationship patterns, and spatial–temporal 3D hand motion trajectory patterns, were built into the proposed model for greater efficiency. Then, a time-independent classification tool was used as a classifier. The main contributions of this study are as follows.

(a)We propose a method for a portable sign language recognition system based on a backhand approach to allow mobility by attaching a 3D small depth sensor to the chest area to detect the skeletons of hands.(b)The proposed method includes a new feature-based recognition approach in terms of the relationship between the hands and key points of the essential body to identify sign words with a similar shape, rotation, and movement, but with different meanings.(c)The developed system can be used with both single and double hands.

The remainder of this article is structured as follows. In Section 1 we present the introduction; in Section 2, we present related works; in Section 3, we present a problem analysis; in Section 4, we provide a system overview; in Section 5, we describe the proposed method; in Section 6, we report on our experiments and results; in Section 7, we discuss the implemented approach, and Section 8 concludes the work.

## 2. Related Works

Many existing works have developed feature extraction techniques [14] to identify similar sign words and letters in a sign language recognition system. They can be divided into two groups. First, the sensor-based group [15,16,17,18,19,20] includes devices worn on the users’ hands, such as gloves, EMG sensors, cables, accelerometers, touch sensors, and flexion sensors. The degree of flexion of the fingers and the fingers’ motion are used as the extracted features. However, with these systems, the user feels uncomfortable due to the complicated equipment, and it is difficult to create a portable system. The second is the vision-based group, which is more popular, and these systems use cameras as the primary device to facilitate motion analysis and interpretation by capturing video sequences. The advantage of a camera is that it eliminates the need for sensors in sensory gloves and reduces system construction costs. The vision-based group consists of 2D and 3D approaches, as shown in Table 1. The 2D method [21,22,23,24,25,26,27,28,29,30,31,32,33] uses video and image data for sign language interpretation. For example, a frequency-based model [21,22] has been proposed which uses a histogram, discrete wavelet transform, and a Gabor filter for extracting features from 2D images. Another proposed approach is the deformable gabarit model [23,24], which uses an active contour and the hand shape as features. Subsequently, the motion-based model [25] was proposed, which uses the motion of the hand as a feature via hand trajectory tracking. Finally, the deep-based model [26,27,28,29,30,31,32,33,34,35] has been proposed, which uses a novel deep-learning-based architecture to learn spatial hierarchies of features automatically and adaptively for 2D images, such as the convolutional neural network model (CNN), the convolutional neural network–long short-term memory model (CNN-LSTM), and recurrent neural networks (RNN). However, the 2D approach has less information than the 3D approach, which makes it hard to solve the problem of similar gestures based on a backhand view. Hence, the 3D approach has been proposed, which can be classified into three groups. The first group is the single-hand group [36,37,38,39,40,41,42], in which a one-hand model-based method is proposed that uses a Leap Motion sensor to detect the finger joints in three dimensions. A feature-based model was used to identify sign letters or sign words, with features such as the angle between fingers; the distance between fingers; finger position; the pitch, roll, and yaw angles of hand; and the trajectory of the finger. Although this method is highly accurate and takes less time to process, and also involves few background problems, this model can only be used with one hand; however, sign language sometimes involves the use of both hands. The second group is the double-hand group [12]. A double-hand-based method was introduced using Leap Motion sensors to solve the problem of similar shapes but different movements and rotations. This method adds a special feature that can distinguish the palm’s rotation angle, such as the pitch, roll, and yaw angles. However, the hand’s rotation is similar in the case of SRM words because these words have similar hand shapes and movements. The third group is the group including both single- and double-hand signs [43,44,45,46,47,48,49], in which both single- and double-hands are used. This group proposed a method to extract features from the hand, such as joint angles, fingertip position, finger direction, and finger velocity. These methods used the 3D model feature extraction method, resulting in high accuracy. More importantly, the backhand approach has been applied [2,3,6,11,12], which is essential for the practical application of the system in daily life. However, these features are hard to use in the case of SRM signs because of the problems with their similar shape, rotation, and movement.

## 3. Problem Analysis

Many sign words have a similar shape, rotation, and movement, and these are called the SRM sign group [13]. Consequently, the features used in the existing work may not be sufficient to distinguish these types of sign words. In the previous research on the backhand view [2,12], a similar problem was solved using rotation-based analysis, such as pitch, roll, and yaw angles. Moreover, another previous work [3,6,11] used position-based feature extraction based on a backhand view. Nevertheless, it would be difficult to apply these features to the SRM sign group, as shown in Figure 1. In the first row of Figure 1, the images in the first column show a single-handed representation of the sign language words “fox” and “fruits”, and the images in the second column show the double-handed representation of the sign language words “brother” and “sister.” The outputs of the rotation-based (including pitch, yaw, and roll angles), motion-based, and shape-based (shape representation of thumb, pinky, and wrist joints in terms of time series) methods are demonstrated in the second, third, and fourth rows, respectively.

As a result, in both the single- and double-hand groups, there are signs with similar shapes, rotation, and movement. However, the position of the hands relative to a part of the body can be used to distinguish words in this group. Therefore, in this paper we present a method based on the spatial–temporal body parts and hand relationship patterns (ST-BHR) as the main feature, using the 3D distance-based Cartesian product, as displayed in Figure 2, which can be expressed as in Equation (5).
(1)$$J_{i,\rho ,t}=\left(x_{i,\rho ,t},y_{i,\rho ,t},z_{i,\rho ,t}\right)$$
where *i* = {1,2,3,…,40}, ρ= {right hand (R), left hand (L)}, and *t* = {1,…,*T*} stand for the total number of finger joints, ρ = R; *i* = {1, 2, 3,…,20} and ρ= L; *i* = {21, 22, 23,…,40}, and the total number of frames, respectively.

According to the solution presented above, $F_{\left(t\right)}$ and $B_{\left(t\right)}$ are the fingertip and palm position set and the key positions of the body part set, as displayed in Equations (2) and (3), respectively. The key position of the selected body part is created by pointing a fingertip of the index finger to the desired locations at the point of the set in Equation (3).
(2)$$F_{\left(t\right)}=\{J_{\left(1,L,t\right)},\;J_{\left(5,L,t\right)},J_{\left(9,L,t\right)},J_{\left(13,L,t\right)},J_{\left(17,L,t\right)},J_{\left(20,L,t\right)},\;J_{\left(21,R,t\right)},\;J_{\left(25,R,t\right)},\;J_{\left(29,R,t\right)},J_{\left(33,R,t\right)},J_{\left(37,R,t\right)},J_{\left(40,R,t\right)}\}$$

Let $F_{\left(t\right)}$, $J_{\left(1,\;L,t\right)}$, L, R, *t* = {1,…,*T*} stand for the set of fingertip and palm positions, fingertip and palm positions, the left hand, the right hand, and the total number of frames, respectively.
(3)$$B_{\left(t\right)}=\left\{S_{1},S_{2},S_{3},S_{4},S_{5},S_{6},S_{7},S_{8}\right\}$$

Here, $B_{\left(t\right)}$, $S_{1}$, $S_{2}$, $S_{3}$, $S_{4}$, $S_{5}$, $S_{6}$, $S_{7},\;S_{8}$ stand for the set of the key points of the body positions, the forehead $\;\left(S_{1}\right)$, the right ear $\left(S_{2}\right)$, the left ear $\left(S_{3}\right)$, the nose ($S_{4}$), the chin ($S_{5}$), the right shoulder $\left(S_{6}\right)$, the left shoulder $\left(S_{7}\right)$, and the chest $\;\mathrm{position}\left(S_{8}\right)$, respectively.

The 3D distance between two points, such as $F_{\left(t\right)}\left(i\right)=\left(x_{\left(i,t\right)},y_{\left(i,t\right)},z_{\left(i,t\right)}\right)$ and $B_{\left(t\right)}\left(k\right)=\left(x_{\left(k,t\right)},y_{\left(k,t\right)},z_{\left(k,t\right)}\right)$ in *xyz*-space, is given by the following generalization of the distance formula in Equation (4).
(4)$$\varphi_{\left(t\right)}\left(F_{\left(t\right)}\left(i\right),B_{\left(t\right)}\left(k\right)\right)=\sqrt{{\left(x_{\left(i,t\right)}-x_{\left(k,t\right)}\right)}^{2}+{\left(y_{\left(i,t\right)}-y_{\left(k,t\right)}\right)}^{2}+{\left(z_{\left(i,t\right)}-z_{\left(k,t\right)}\right)}^{2}}$$
where $\varphi_{\left(t\right)}$,$F_{\left(t\right)}$, $B_{\left(t\right)}$, *x*, *y*, *z*, *i* = {1,2,3…,12}, *k* = {1,2,3…,8} stand for the 3D distance, the set of fingertip and palm positions, the set of the key positions of the body parts, the *x*-axis, the *y*-axis, the *z*-axis, the total of set $F_{\left(t\right)}$, and the total of set $B_{\left(t\right)}$, respectively.

The proposed method, 3D distance-based features based on Cartesian products, is used as the feature extraction method, as shown in Figure 3 and Equation (5), in which the output of the Cartesian product is normalized before feeding it into a classifier.
(5)$$H_{1}{}_{\left(t\right)}=Norm\left[F_{\left(t\right)}\times B_{\left(t\right)}\right]=Norm\left\{\varphi_{jk}\left(J_{\left(i,h,t\right)},S_{n}\right)|\left(J_{\left(i,h,t\right)}\in F_{\left(t\right)}\right)\wedge \left(S_{n}\in B_{\left(t\right)}\right)\right\}$$
where $H_{1}{}_{\left(t\right)}$, $F_{\left(t\right)},\;B_{\left(t\right)},\varphi_{jk}$, $J_{\left(i,h,t\right)}$, and Norm indicate the Cartesian of the set of $F_{\left(t\right)}$ and $B_{\left(t\right)}$, the set of fingertip and palm positions, the set of the key positions of the body parts, the 3D distance, the finger joints, and min–max normalization, respectively.

This feature represents spatial–temporal body parts and hand relationship patterns based on the 3D distance, normalized via min–max normalization [50], as shown in Figure 4. As a result, the words “fox” and “fruit” show different patterns in terms of the time series of the video, represented by a red line between the circle and the rectangle, which stands for the normalized distance of the words “fox” and “fruit”, respectively.

## 4. System Overview

Hearing-impaired people usually communicate through sign language, whereas non-hearing-impaired people normally use speech based on a natural language. An approach that allows these two groups of people to communicate via a mobile device that converts sign language to natural language and vice versa in real-time is illustrated in Figure 5. When hearing-impaired people speak in front of hearing-impaired and non-hearing-impaired people, hearing-impaired people can understand the meaning based on the signs, while simultaneously, non-hearing-impaired people can listen to the speech. In this scenario, a mobile device is installed on a body part of a hearing-impaired signer, such as the chest, face, and head, and functionally converts the video of signs into speech for non-hearing-impaired people to understand and vice versa.

### 4.1. Hardware Unit

The Leap Motion sensor [51] or 3D sensor is set on the signer’s chest so that it automatically moves along the moving path of the signer and always detects hand signs in the same view or the backhand view, as illustrated in Figure 6. In this paper, we focus on translating sign words and showing them as text, presented in a red rectangle. In addition, the interaction zone of the sensor extends from 10 cm to 80 cm, and a 140° × 120° typical field of view. Therefore, if the sensor is in the normal plane with no tilt, an interaction zone is unable to detect the selected body point. The interaction zone can be designed as shown in Figure 6.
(6)$$\theta_{\mathrm{base}\;}=\theta_{1}+\delta ;\;h_{1}$$

Therefore, $\theta_{base}$,$\;\theta_{3\;}=120{}^{\circ}$, $\theta_{1\;}$, *δ*, and $h_{1}$ stand for the tilt angle of the sensor, the vertical angle of the 3D sensor, an angle, a bias angle, and the sensor’s position in terms of the chest area determined by the height from the shoulder to the chest, respectively.

### 4.2. Software Unit

The software unit is divided into three parts, as illustrated schematically in Figure 7. First, the preprocessing step is performed, containing the data mining and analysis processes, such as the receival of 3D skeleton hand data from the 3D depth sensor and the 3D body input, obtained from individual calibration, as described in Section 5.1. Second, the feature extraction process consists of four features: the spatial–temporal body parts and hand relationship patterns as the main feature, the spatial–temporal finger joint angle patterns, the spatial–temporal double-hand relationship patterns, and the spatial–temporal 3D hand motion trajectory patterns based on PCA, as reported in Section 5.2. Thirdly, the classification method, specifically, the two-layer BiLSTM neural network, can be used to deal with time-independent data, as described in Section 5.3. Otherwise, the system returns to the start of the loop, with new finger joint positions being obtained by the 3D sensor.

## 5. Proposed Method

In this section, we present the preprocessing, feature extraction, and classification technique, which is shown schematically in Figure 8.

### 5.1. Preprocessing Technique

The preprocessing technique consists of two parts: 3D skeleton joint data received from the data acquisition device describe the 3D skeleton hand. The calibration technique presents the 3D points of interest of the body area obtained via the calibration using the tip of the index finger as a pointer.

#### 5.1.1. 3D Skeleton Joint Data

The 3D sensor is an optical hand tracking module that captures the movements of hands, such as the 3D position and direction. In the case of a single hand, the zero padding technique [2] is used to replace the absent of finger joints of the second hand; it is assumed to be zero. Then, the total number of finger joints is expressed in the *t*-frame, as demonstrated in Equation (1).

#### 5.1.2. Calibration Technique

The implementation of a calibration technique is an important step before using the system, and this must be a calibration for an individual person. The result of this step is the three-dimensional positioning of key points of the body by placing the tip of the index finger on the desired locations at the points of set $B_{\left(t\right)}$ and obtaining the fingertip’s position using the Leap Motion sensor as a reference position point, as shown in Figure 2 and Equation (7).
(7)$$P_{B_{\left(t\right)}\left(i\right)}(x,\;y,\;z)=T_{B_{\left(t\right)}\left(i\right)};\;B_{\left(t\right)}=\{S_{1},\;S_{2},\;S_{3},\;S_{4},\;S_{5},\;S_{6},\;S_{7},\;S_{8}\}$$
where $P_{B_{\left(t\right)}\left(i\right)}$, $T_{B_{\left(t\right)}\left(i\right)},\;B_{\left(t\right)}$, *i* = 1,…,7 stand for the new 3D positions of key points of the body, the 3D position created by pointing a tip of the index finger to the desired locations at the point of set $B_{\left(t\right)}$ according to Equation (2), the set of the key points of the body positions, and the total number of selected body points, respectively.

### 5.2. Feature Extraction

In this section we use the spatial–temporal body parts and hand relationship patterns as the main feature to identify signed words with similar shapes, rotation, and movement. Moreover, in this study we have added more features to make the system more efficient because, in addition to solving the problems mentioned above, it can also be used with other samples [37], since in this work we have applied both single- and double-handed approaches. The case of double-handed signs involves the relationship of the left and the right hand. Therefore, we extract four features: the spatial–temporal body parts and hand relationship patterns, the spatial–temporal finger joint angle patterns, the spatial–temporal double-hand relationship patterns, and the spatial–temporal 3D hand motion trajectory patterns. These are proposed in Section 5.2.1, Section 5.2.2, Section 5.2.3 and Section 5.2.4, respectively.

#### 5.2.1. Spatial–Temporal Body Parts and Hand Relationship Patterns

The spatial–temporal body parts and hand relationship patterns are the feature that determines the relationship between the position of the left and right hands with the selected points on the body to solve the problem of words which have similar shapes and movements, but different finger positions. However, information on the relationship of both hands and the selected 3D position points on the body can be used to solve this problem by calculating the distance between each 3D point, as shown in Figure 2. All distance is normalized to store a set of patterns in terms of spatial–temporal data, as shown in Equation (5).

#### 5.2.2. Spatial–Temporal Finger Joint Angle Patterns

We have proposed the use of this feature due to the similar shapes used in sign words [2,12]. There are two kinds of features, as indicated in Figure 9. Firstly, finger joint angles are used to find the angles between the finger joints of the same finger and the angles between two adjacent fingertips, determined as follows in Equation (8). This feature can be used to characterize the shape of the hand. Secondly, pitch, yaw, and roll angles indicate the palm orientation, enabling us to obtain the pitch (ρ) (angle of the *x*-axis), yaw (ϕ) (angle of the *y*-axis), and roll (∄) (angle of the *z*-axis), as shown in Equations (9)–(11), respectively.
(8)$$\theta =\arccos\left(\frac{\vec{M}\cdot \vec{N}}{∥\vec{M}∥\cdot ∥\vec{N}∥}\right)$$
where θ is the angle between $\vec{M}$ and $\vec{N},$ which indicate the 3D finger joint positions.
(9)$$\rho =\arctan\left(\frac{A_{X}}{\sqrt{A_{Y}^{2}+A_{Z}^{2}}}\right)$$
(10)$$\phi =\arctan\left(\frac{A_{Y}}{\sqrt{A_{X}^{2}+A_{Z}^{2}}}\right)$$
(11)$$∄=\arctan\left(\frac{\sqrt{A_{X}^{2}+A_{Y}^{2}}}{A_{Z}}\right)$$
where ρ, ϕ, $∄,{\;A}_{X}\;$ $A_{Y}$, and $A_{Z}$ are the pitch angle, yaw angle, and roll angle of the palm orientation, the *x*-axis, the *y*-axis, and the *z*-axis, respectively.

In the final step, the pitch, yaw, and roll angles of the palm joint and the finger joint angles of the consecutive joint in the same finger must be collected in the set, in which all data are normalized to store a set of patterns in terms of the spatial–temporal data in $H_{2}\left(t\right)$, as shown in Equation (12).
(12)$$H_{2}\left(t\right)=Norm\left[\left\{\theta_{1}\left(t\right),\theta_{2}\left(t\right),\theta_{3}\left(t\right),\ldots ,\theta_{n}\left(t\right),\;\rho_{R}\left(t\right),\phi_{R}\left(t\right),{∄}_{R}\left(t\right),\;\rho_{L}\left(t\right),\phi_{L}\left(t\right),{∄}_{L}\left(t\right)\right\}\right]$$
where Norm,$\;t=\left\{1,\ldots ,T\right\}$, *R*, *L*, $\rho \left(t\right)$, $\phi \left(t\right)$, and ∄(*t*) are the min–max normalization, the total number of frames, the right hand, the left hand, the pitch angle, the yaw angle, and the roll angle, respectively.

#### 5.2.3. Spatial–Temporal Double-Hand Relationship Patterns

Due to the similar shapes of sign words [6], the spatial–temporal double-hand relationship patterns are proposed based on 3D Euclidean distance, as demonstrated in Figure 10 and Equation (13). This feature describes the relationship of the left and right hands to solve the problem of words or letters signed with similar movements. Therefore, all 3D distances are collected in the set; then, the selected distance-based data are normalized to store a set of $\;H_{3}\left(t\right)$, which is the term of the spatial–temporal data.
(13)$$H_{3}\left(t\right)=Norm\left[\left\{U_{1}\left(t\right),\;U_{2}\left(t\right),U_{3}\left(t\right),\ldots ,U_{n}\left(t\right)\right\}\right]$$
where $U_{n}\left(t\right)$, Norm,$\;t=\left\{1,\ldots ,T\right\}$, and *n*= {1, 2, 3, …, 20} are the distance between two points in *xyz*-space, min–max normalization, the total number of frames, and the total number of points, respectively.

#### 5.2.4. Spatial–Temporal 3D Hand Motion Trajectory Patterns

Due to the similar shapes used in different signs [3], the concept of the spatial–temporal 3D hand motion trajectory patterns are presented to extract the movement trajectory of the finger joints, in which the 3D positions of the joints are collected in the set, as shown in Figure 11. Then, principal component analysis (PCA) [52] is used to reduce the dimensions of the data set to one dimension. Finally, the selected data are normalized to store a set of $H_{4}\left(t\right)$, the term of the spatial–temporal data, as shown in Equation (14).
(14)$$H_{4}\left(t\right)=Norm[\mathrm{PCA}\{(T_{1}\left(t\right)),(T_{2}\left(t\right)),(T_{3}\left(t\right)),\ldots ,(T_{n}\left(t\right))\}]$$
where $T_{n}\left(t\right)$, Norm, PCA, and $t=\left\{1,\ldots ,T\right\}\;$ are the tip and palm positions in the time series of *xyz*-space, min–max normalization, principal component analysis, and the total number of frames, respectively.

In the final stage of feature extraction, the four features consist of the spatial–temporal body parts and hand relationship patterns ($H_{1}\left(t\right)$), the spatial–temporal finger joint angle patterns ($H_{2}\left(t\right)$), the spatial–temporal double-hand relationship patterns ($H_{3}\left(t\right)$), and the spatial–temporal 3D hand motion trajectory patterns ($H_{4}\left(t\right)$). These features are concatenated into one-dimensional data by means of a concatenation technique, of which the equations are shown in Equation (15). The final feature extraction in terms of spatial temporal patterns ($X\left(t\right)$) is the input of a stacked BiLSTM in the classification process.
(15)$$X\left(t\right)=\left\{\left[\left[\left[H_{1}\left(t\right)∥H_{2}\left(t\right)\right]∥H_{3}\left(t\right)\right]∥H_{4}\left(t\right)\right]\right\}$$
where $H_{1}\left(t\right)$, $H_{2}\left(t\right),\;H_{3}\left(t\right)$, $H_{4}\left(t\right),∥,\;$*t* = {1,…,*T*} stand for the spatial–temporal body parts and hand relationship patterns, the spatial–temporal finger joint angle patterns, the spatial–temporal double-hand relationship patterns, the spatial–temporal 3D hand motion trajectory patterns, the concatenation operator, and the total number of frames, respectively.

### 5.3. Classification

A deep learning algorithm, a recurrent neural network (RNN), is applied for the analysis of data in the form of a serial sequence, such as a video (a series of images) or text (a sequence of words). However, RNNs exhibit a vanishing and exploding gradient problem [53], which results in poor performance when dealing with long sequences. Therefore, bidirectional long short-term memory (BiLSTM) [54] is used for solving long sequences because it is possible to choose which data should be remembered or eliminated. Recent experimental works [3,12,37,55] have demonstrated that the BiLSTM network outperformed various models, such as the standard CNN, SVM, RNN, ARMA, and ARIMA. The use of a single LSTM network for sign word recognition led to low accuracy and overfitting, especially when learning complex sign sequences. To address this problem, stacking more than one BiLSTM unit, as in [2,43], enhances performance in the recognition of sign words. Therefore, inspired by these works, we designed our BiLSTM architecture using two BiLSTM units, as shown in Figure 12. This two-unit BiLSTM architecture allows us to achieve the high-level sequential modeling of the selected features.

The structure of the designed two-layer BiLSTM model consists of the input, BiLSTM hidden neurons, dropout, and classification layers, as shown in Figure 12. The input layer contains time series data, as demonstrated in Equation (15). Then, the BiLSTM hidden layer consists of four main gates: the forget gate, the input gate, the input modulation gate, and the output gate. The forget gate ($f_{t+1}$) controls the flow of information to forget or keep the previous state ($C_{t}$). The input data ($x_{n\left(t+1\right)}$), previous hidden state ($h1_{t}$), bias $\;(b_{f})$, and the sigmoid function (σ) are used for making a decision, as shown in Equation (16). If the forget gate is set to 0, it decides to forget the previous state, but if the forget gate is set to 1, it keeps the previous state. However, the input gate ($i_{t+1}$) is used to decide which information of the input ($x_{n\left(t+1\right)})$,$\;\mathrm{sigmoid}\;\mathrm{function}\;\left(\sigma \right),$ and $\;h1_{t}$ should be passed to update the cell state, as demonstrated in Equation (17). Equation (18) expresses the cell states generated from the updated cell state in terms of the forget gate ($f_{t+1}$), the input gate ($i_{t+1}$), and the previous state ($C_{t}$), respectively. Then, the output gate ($o_{t+1}$) decides the value for the next sequence $h1_{t+1}$, as shown in Equations (19) and (20). After that, a dropout layer is proposed to prevent overfitting by randomly turning off nodes in the network [56]. Based on previous works [57], the dropout layer was compared with each dropout level rate, resulting in dropout value of 0.2, which showed the best performance. Therefore, a dropout value of 0.2 was applied in this study. Lastly, the Softmax function is applied in the classification layer. The Softmax function is a form of logistic regression that normalizes an input value into vector values, following a probability distribution, in which the output values are in the range of [0,1].
(16)$$\;f_{t+1}=\sigma \left(x_{n\left(t+1\right)}W_{fx}+h1_{t}W_{fh}+b_{f}\right)$$
(17)$$\;i_{t+1}=\sigma \left(x_{n\left(t+1\right)}W_{ix}+h1_{t}W_{ih}+b_{i}\right)$$
(18)$$C_{t+1}=(C_{t}\times f_{t+1})+(i_{t+1}\times tanh\left(x_{n\left(t+1\right)}W_{\Theta x}+h1_{t}W_{\Theta h}+b_{\Theta })\right)$$
(19)$$\;o_{t+1}=\sigma \left(x_{n\left(t+1\right)}W_{ox}+h1_{t}W_{oh}+b_{o}\right)$$
(20)$$\;h1_{t+1}=o_{t+1}\times tan$$
where $\;f_{t+1}$, $\;i_{t+1}$, $C_{t+1}$, $\;o_{t+1}$, $\;h1_{t+1}$, $x_{n\left(t+1\right)}$, σ, $h1_{t}$, $C_{t}$, tan*h*, $W\in R^{U\times V}$, and $b\in R^{V}$ stand for the forget gate vector, the input gate vector, the cell input vector, the output gate vector, the hidden state vector, the input data, the sigmoid function, the previously hidden state vector, the previous cell state vector, the hyperbolic tangent function, the weight matrices, in which the superscripts U  and V  refer to the number of input features and the number of hidden units, and the bias vector parameters, in which the superscripts V refer to the number of hidden units, respectively.

## 6. Experiments and Results

Figure 13 shows the experimental setup used for training and testing the proposed idea. It consisted of a sensor that was set up in the chest area and a computer system in which a laptop was used to collect the output from the sensor.

### 6.1. Experiments

This section consists of three parts: firstly, the dataset, which describes the dataset used in the experiment; the type of dataset used; and the dataset from the existing research that was used to compare the efficiency of the system. Secondly, the configuration parameters describe the parameter settings of the sign language recognition system and the classification model. Finally, we present an evaluation of the classification model and give details regarding the system’s validity.

#### 6.1.1. Dataset

The American Sign Language dataset includes signed words and signed letters which are currently used in sign language recognition. Most are forehand methods, but in this study, we propose a backhand approach. Therefore, it is compulsory to create a new dataset due to insufficient information. The created datasets are divided into two types—single-handed and double-handed datasets of signed words, with 36 words, giving a total of 72 words. The datasets were collected by ten deaf and hard-of-hearing people, with each person collecting each word ten times so that both types would contain 7200 samples. This method uses the backhand approach; therefore, the sensor must be set on the chest and can be used in both day and night, as shown in Figure 2. In addition, this method has been tested on signed letters (the letters A–Z) that exhibited the problem of similar signed letters, using the datasets from [3], which presented a backhand dataset of 5200 samples. In the same way, this method was also tested with the dataset of signed words from [12], in which they collected 40 double-hand dynamic ASL words, giving a total of 4000 samples of similar signed words. Therefore, the total dataset contained 16,400 samples, as listed in Table 2. The protocol of *k*-fold cross validation was applied with these datasets.

#### 6.1.2. Configuration Parameter

The configuration parameters can be categorized into two parts. The first part, the hardware specifications, describe the computer system and the Leap Motion sensor [51] details, as given in Table 3. In the second part, the classification parameter settings are used to configure the parameters of the BiLSTM model, as demonstrated in Table 4.

#### 6.1.3. Evaluation of the Classification Model

Accuracy [12] is described as a measure of correct predictions. Accuracy is given by Equation (21). The standard deviation is used to measure the amount of variation of a set of values, as shown in Equation (22). Moreover, the error, precision, recall, and F1-score [12] are applied for this model.
(21)$$\;Accuracy\;(ACC)\;=\frac{\Upsilon_{1}+\Upsilon_{2}}{\Upsilon_{1}+\Upsilon_{2}+\phi_{1}+\phi_{2}}$$
where $\Upsilon_{1}$, $\Upsilon_{2}$, $\phi_{1}$, and $\phi_{2}$ denote the true positive (correctly identified), true negative (correctly rejected), false positive (incorrectly identified), and false negative (incorrectly rejected) values, respectively.

(22)$$SD=\sqrt{\frac{\sum_{i=1}^{n}{\left(x_{i}-\Upsilon_{3}\right)}^{2}}{n}}$$ where *SD*, $\Upsilon_{3}$, $x_{i}$, and n stand for the standard deviation, the mean of all samples, each sample, and the total number of samples, respectively.

### 6.2. Results

The performance comparison in the task of signed-letter recognition (letters A–Z) using the experimental data set in [3] is demonstrated in Table 5. The overall evaluation results were obtained using measures of accuracy, error, precision, recall, F1-score, and standard deviation (SD), respectively. The recognition results of 26 signed letters using the proposed method were compared with the conventional method presented in [37] based on a feature-based model, specifically a finger-motion-based forehand view, and the method using a trajectory-based backhand view [3]. The results of the proposed method demonstrated that our proposed features improved the accuracy rate in signed-letter recognition (letters A–Z) by about 1.27%. The other metrics under consideration were not available in [3,37]; thus, these columns contain no data.

Moreover, the performance comparison of the experimental results obtained for 40 double-hand dynamic ASL words from [12] using our proposed method is shown in Table 6. Using the proposed method, the accuracy rate was improved by 0.54 % compared to the conventional method [12] through the use of shape, motion, position, and angle-based features. As shown in Table 6, the conventional method [12] achieved an accuracy of 97.98%, whereas our proposed method increased the accuracy by 0.54%, with an overall accuracy of 98.52%, an error of 1.48%, a precision of 98.56%, a recall of 98.52%, an F1-score of 98.54%, and an SD of 0.22%. The overall accuracy for 72 American Sign Language (ASL) words was 96.99%. The performance comparison of the experimental results in the recognition of signed-words (72 words, including single- and double-handed ASL words) is shown in Table 7. We performed an ablation test, to demonstrate the significance of the proposed features over those used in previous works.

#### Ablation Test

We conducted ablation tests to determine the significance of each feature in the proposed models. The feature combination used in this method consists of the spatial–temporal body parts and hand relationship patterns ($H_{1}$), the spatial–temporal finger joint angle patterns ($H_{2}$), the spatial–temporal double-hand relationship patterns ($H_{3}$), and the spatial–temporal 3D hand motion trajectory patterns ($H_{4}$). This method of feature extraction for signed-letter recognition evaluates different feature combinations among all the features, as shown in Table 8. We evaluated the $1^{st}$($H_{2}$ + $H_{3}$), $2^{nd}$($H_{2}$ + $H_{3}$ + $H_{4}$), and $3^{rd}$($H_{1}$ + $H_{2}$ + $H_{3}$ + $H_{4}$) combinations, and found that the $3^{rd}$($H_{1}$ + $H_{2}$ + $H_{3}$ + $H_{4}$) combination was the best combination of features for signed-letter recognition (letters A–Z), achieving an accuracy of 97.34%, a precision of 97.39%, a recall of 97.34%, an F1-score of 97.36%, and an SD of 0.26%. However, the results showed that the use of the $\left(H_{1},\;H_{4}\right)$ features can improve the accuracy.

In our performance comparison, we conducted an ablation test related to signed-word recognition, and evaluated different feature combinations, as shown in Table 9. The feature combination consisted of the spatial–temporal body parts and hand relationship patterns ($H_{1}$), the spatial–temporal finger joint angle patterns ($H_{2}$), the spatial–temporal double-hand relationship patterns ($H_{3}$), and the spatial–temporal 3D hand motion trajectory patterns ($H_{4}$). There are three different feature combinations models: the $1^{st}$($H_{2}$ + $H_{3})$, $2^{nd}$($H_{2}$ + $H_{3}$ + $H_{4}$), and $3^{rd}$($H_{1}$ + $H_{2}$ + $H_{3}$ + $H_{4}$) combinations. The results showed that the proposed model provided the best performance, with an accuracy of 98.52%, a precision of 98.56%, a recall of 98.52%, an F1-score of 98.54%, and an SD of 0.22%.

The experimental results for 72 sign words, including single- and double-handed words, based on a five-fold cross-validation [58], which is one of the most regularly used model evaluation methods, are shown in Table 10 and Table 11, and Figure 14. In Table 10, the group (G.) in the table indicates a pair of words with similar signs. The error column indicates other misspelled words. For example, in Table 11, group (G.) 13 of the word “father” displayed an error in relation to the word “mother” of 2.6%. As a result, this indicates that the system has high accuracy. 

## 7. Discussion

In the case of using a single hand, there is some hand occlusion by the palm, which obstructs the view of the sensor, resulting in an incorrect predictive finger position and a missing position when the finger is occluded. For example, the word “spit” in the third row causes occlusion by the palm, as shown in Figure 15. Unfortunately, the predicted finger joint position in the second row is similar to the position of the word “grandmother” in the first row, causing the determined meaning to be incorrect. For future solutions, prior and post-information may be used to predict the lost position of the finger.

In the second case of single- and double-hand words, some hand occlusion problems are caused by fingers. For example, in Figure 16, the position of the word “are” in the third row, which is derived from the sensor, has the wrong position in the second row. Regrettably, this wrong position is similar to the word “true”, shown in the first row, thus causing a misclassification problem. Moreover, in terms of double-hand words, Figure 17 shows an instance in which the error position of the word “keep” in the second row is similar to the word “sister” in the first row.

In addition, since the Leap Motion sensor is mounted on the chest, which limits the scope of the area in which the hand can be detected, there is a problem with some signed words such as “introduce”, for which the hand position is sometimes difficult with the interaction zone. The solution for this problem is to increase an interaction zone by using two Leap Motion sensors.

## 8. Conclusions

When using a backhand approach, some signed words have a similar shape, rotation, and hand movement, but different hand positions; thus, detection systems suffer from misclassification errors, which results in lower accuracy. Therefore, in this study we propose the use of the spatial–temporal body parts and hand relationship patterns (ST-BHR) as the main feature, in which the set of the 3D positions of the finger joints and the set of the key positions of the body parts are applied, measuring the Euclidean distances based on a Cartesian product to derive a series of 3D distance-based features. Then, the bidirectional long short-term memory method is used as a classifier for time-independent data. The performance of the method was evaluated using 72 sign words from 10 participants, using both single- and double-handed words, and the accuracy was found to be approximately 96.99%. The method was further developed on 26 ASL sign letters and 40 double-hand dynamic ASL word datasets, improving upon the conventional method by 1.27% and 0.54%, respectively.

The evaluation results demonstrate that the use of ST-BHR features minimized the problems associated with the SRM sign group. Therefore, the proposed method can allow people who are hard-of-hearing to access the recognition system at their convenience. However, a major limitation of this work is the limited interactive space used for hand and face position detection. In future works, two Leap Motion sensors can be used to expand the interaction zone. Moreover, real social situations [59,60] will be examined in future work, introducing factors such as vibration [61] from driving and other means of transportation which would affect the position of the camera sensor. Therefore, this issue will be a challenging topic in future.

## Figures and Tables

**Figure 1 sensors-22-04554-f001:**
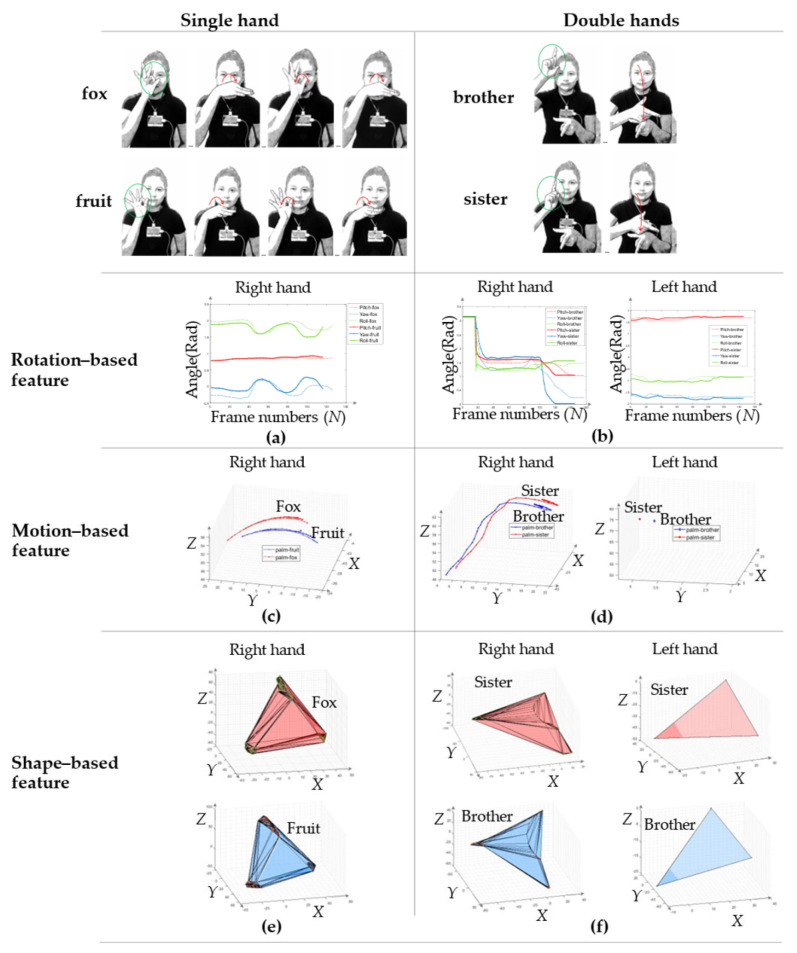
Problem analysis of different features used for similar sign words of (**a**) Rotation representation of hand of single hand; (**b**) Rotation representation of hand of double hands; (**c**) Motion representation of hand of single hand; (**d**) Motion representation of hand of double hands; (**e**) Shape representation of thumb, pinky, and wrist finger on time series of single hand; (**f**) Shape representation of thumb, pinky, and wrist finger on time series of double hands.

**Figure 2 sensors-22-04554-f002:**
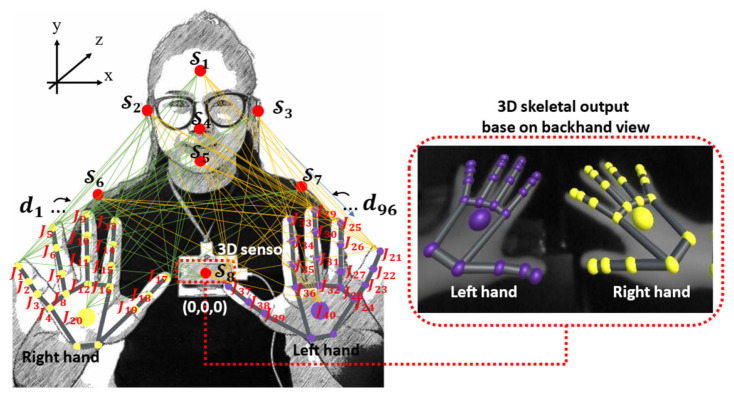
Spatial–temporal body parts and hand relationship patterns (ST-BHR).

**Figure 3 sensors-22-04554-f003:**
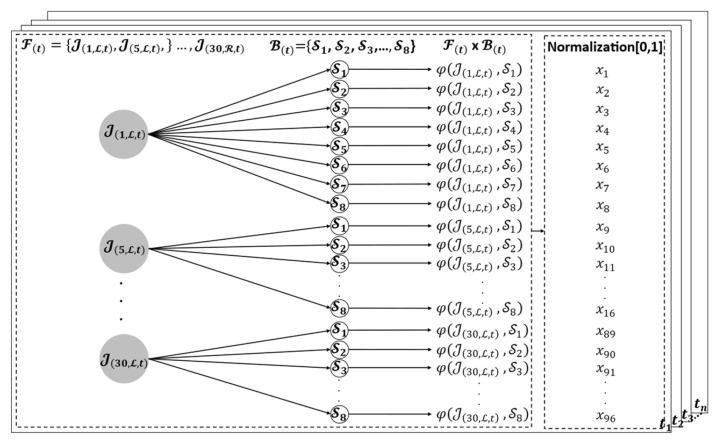
Distance-based features based on Cartesian products.

**Figure 4 sensors-22-04554-f004:**
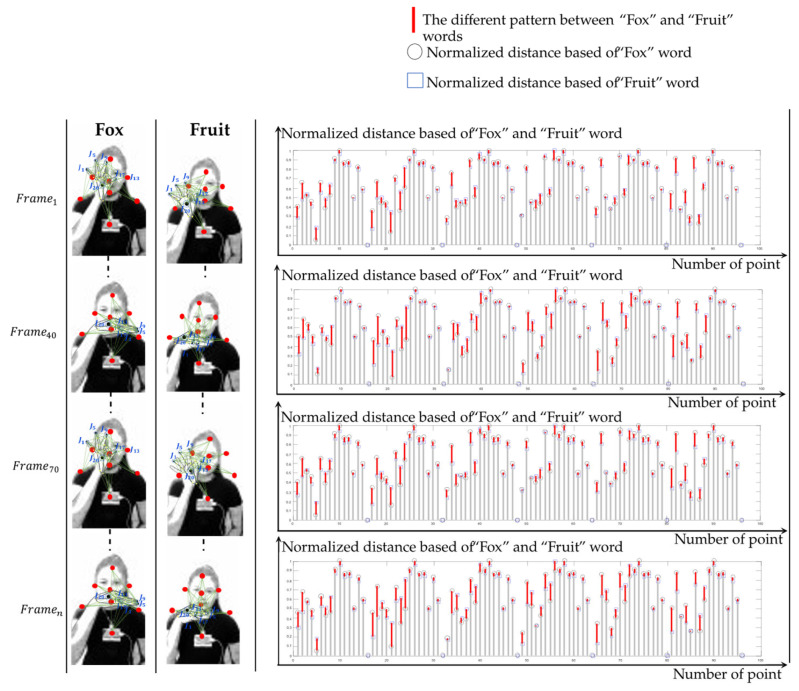
Comparison of similarly signed words using $H_{1}{}_{\left(t\right)}$.

**Figure 5 sensors-22-04554-f005:**
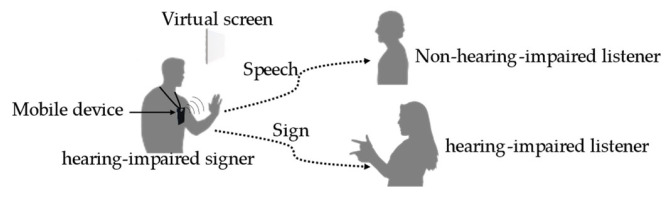
Scenario of the use of a sign-language interpretation device.

**Figure 6 sensors-22-04554-f006:**
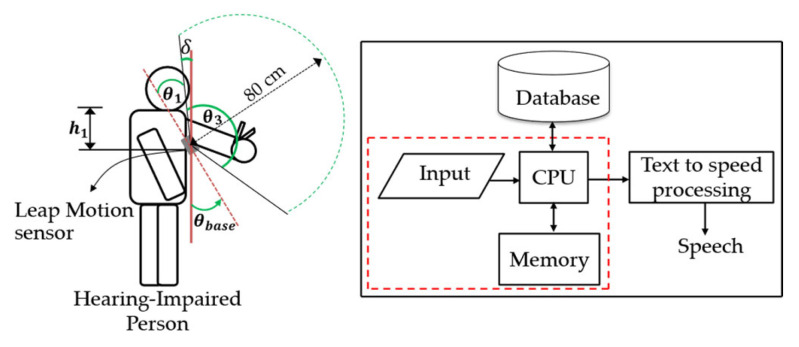
Hardware system.

**Figure 7 sensors-22-04554-f007:**
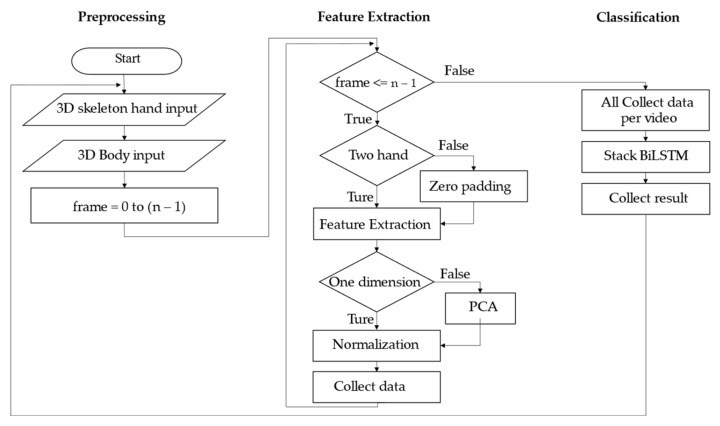
Flowchart of the proposed method.

**Figure 8 sensors-22-04554-f008:**
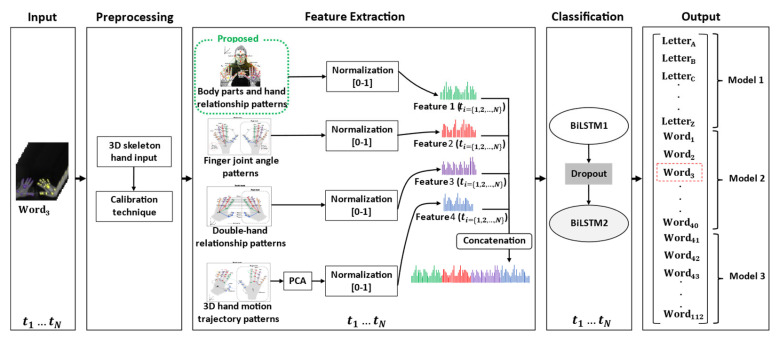
Proposed method.

**Figure 9 sensors-22-04554-f009:**
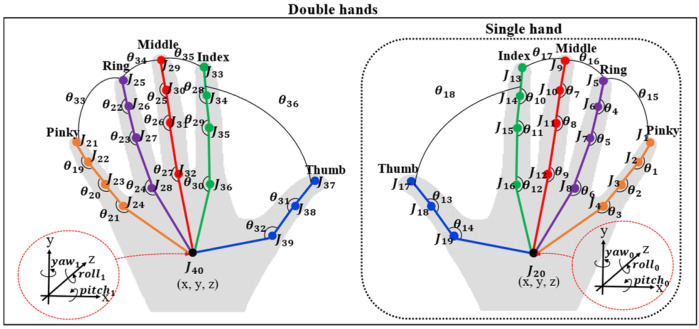
Spatial–temporal finger joint angle patterns.

**Figure 10 sensors-22-04554-f010:**
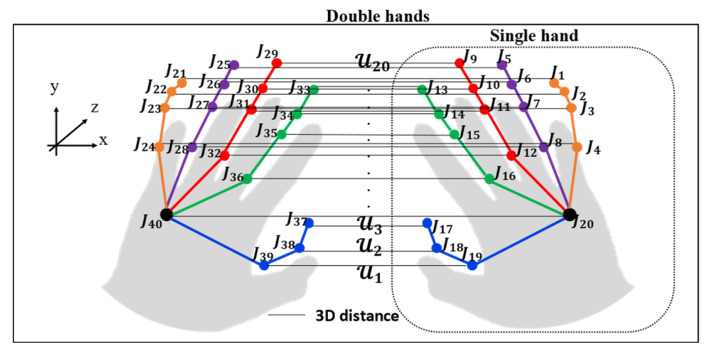
Double-hand relationship patterns for single and double hands.

**Figure 11 sensors-22-04554-f011:**
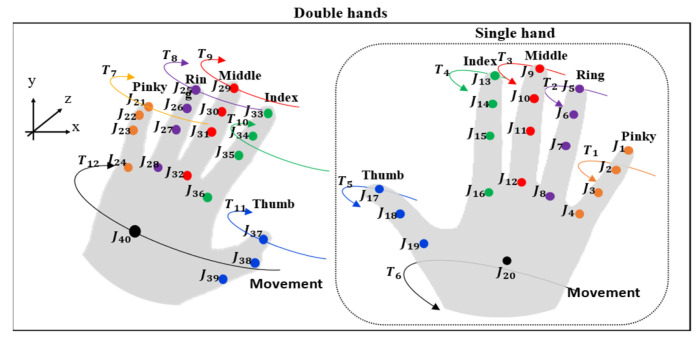
3D hand motion trajectory patterns for single and double hands.

**Figure 12 sensors-22-04554-f012:**
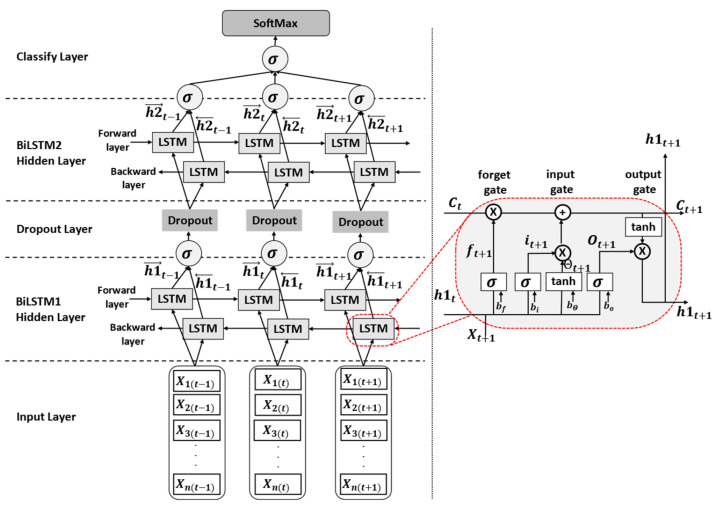
Two-layer BiLSTM neural network.

**Figure 13 sensors-22-04554-f013:**
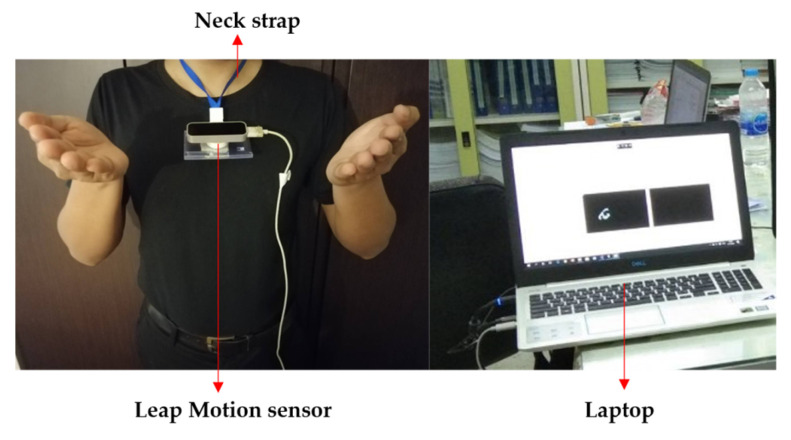
Photograph of the experimental setup.

**Figure 14 sensors-22-04554-f014:**
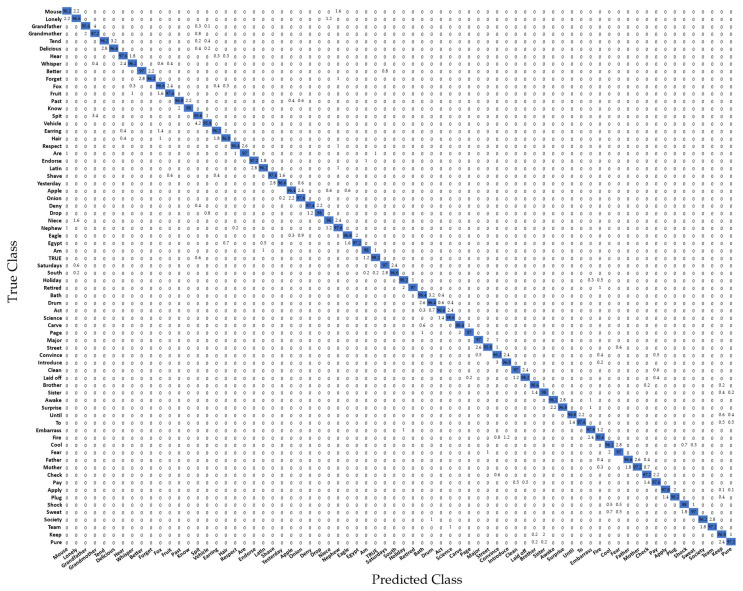
Confusion matrix of the recognition performance of 72 American Sign Language (ASL) words.

**Figure 15 sensors-22-04554-f015:**
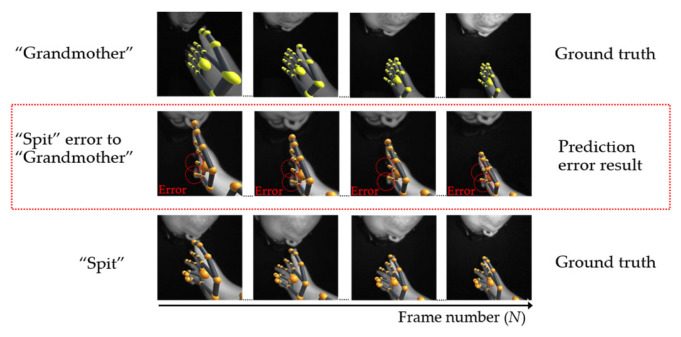
Example of an error caused by the palm.

**Figure 16 sensors-22-04554-f016:**
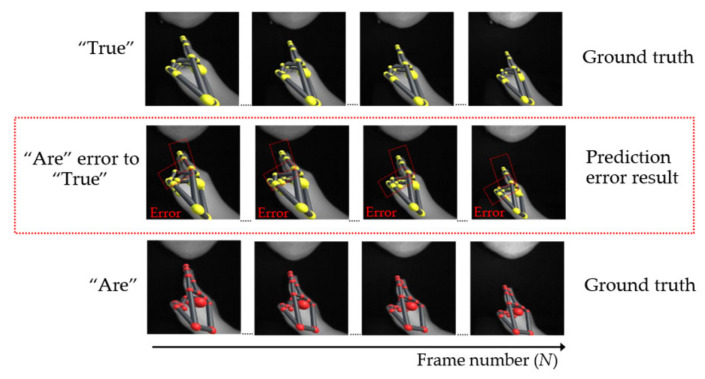
Example of an error caused by a finger.

**Figure 17 sensors-22-04554-f017:**
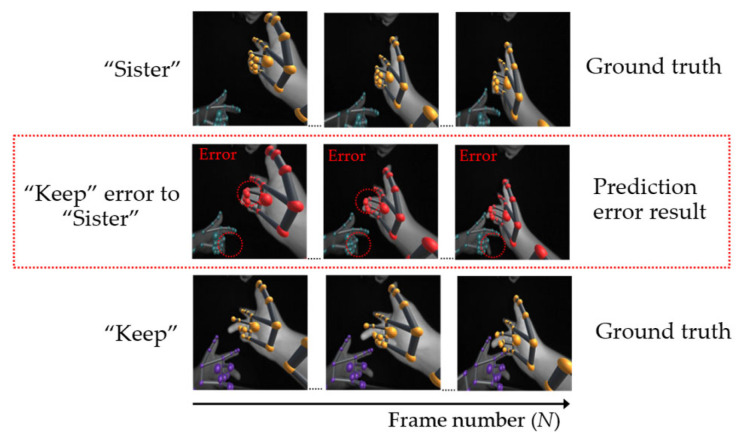
Example of a double-hand word error caused by a finger.

**Table 1 sensors-22-04554-t001:** Vision-based methods of automatic sign language recognition.

References	Methodology	Acquisition Mode	Results (%)	Forehand/ Backhand	Limitation
1. 2D approach					
[22]	24 ASL letters + DWT Gabor filter + KNN	Image	96.70	Forehand	Fails to track SRM sign group
[23]	24 ASL letters + Contour-based + ANN	Image	79.58	Forehand	Fails to track SRM sign group
[24]	100 ASL words + HOG + HMM	Video	98.90	Forehand	Fails to track SRM sign group
[26]	27 hand gestures + Lightweight CNN model	Image	97.25	Forehand	Fails to track SRM sign group
[27]	20 ASL words + CNN model + SVM	Video	97.28	Forehand	Fails to track SRM sign group
[28]	36 ASL letters + CNN model	Image	90.26	Forehand	Fails to track SRM sign group
2. 3D approach					
2.1 Single-hand model					
[36]	26 ASL letters + 30 feature-based + LSTM	Video	91.82	Forehand	Fails to track SRM sign group
[37]	26ASL letters + Position-based + DNN	Video	93.81	Forehand	Fails to track SRM sign group
[38]	26 ASL letters + 56 feature-based + HMM	Video	86.10	Forehand	Fails to track SRM sign group
[39]	12 ASL words + Position-based + D-LSTM	Video	90.00	Forehand	Fails to track SRM sign group
[40]	26 ASL letters + Distance-based + ANN	Video	96.15	Forehand	Fails to track SRM sign group
[41]	26 ASL letters + Distance-based + GBM	Image	87.60	Forehand	Fails to track SRM sign group
[3]	26 ASL letters + Trajectory-based + LSTM	Video	96.07	Backhand	Fails to track SRM sign group
2.2 Double-hand model					
[12]	40 words + Motion and angle-based + BiLSTM	Video	97.98	Backhand	Fails to track SRM sign group
3. Multi-single- and double-hand models					
[43]	30 words + Angle-based + LSTM	Video	96.00	Forehand	Fails to track SRM sign group
[45]	18 words + CNN model	Video	82.55	Forehand	Fails to track SRM sign group
[46]	49 words + Hand skeletal + FV + SVM	Video	86.86	Forehand	Fails to track SRM sign group
[48]	50 words + Position-based + BiLSTM-NN	Video	94.55	Forehand	Fails to track SRM sign group
[6]	56 words + Trajectory-based + HB-RNN	Video	94.50	Backhand	Fails to track SRM sign group
[2]	57 words + Angle-based + FFV-BiLSTM	Video	98.60	Backhand	Fails to track SRM sign group

**Table 2 sensors-22-04554-t002:** Dataset description.

Datasets	No. of Participants	Frequency (Times/Word)	No. of Samples
36 single-hand ASL words (Created by author)	10	10	3600
36 double-hand ASL words (Created by author)	10	10	3600
26 signed letters (A–Z letters) by [3]	10	20	5200
40 double-hand ASL words by [12]	10	10	4000
Total samples			16,400

**Table 3 sensors-22-04554-t003:** Hardware specifications.

Systems	Specification
Computer system	Dell G3 Gaming w56691425TH
	CPU: Intel Core i7-8750H
	GPU: NVidia GeForce GTX 1050Ti
	Memory Size: 8 GB DDR4
Leap Motion sensor	Video: 120 frames per second
	Infrared camera: 2 cameras
	Pixel: 640 × 240
	Interaction zone: 80 cm
	FOV: 150 × 120 degrees
	Accuracy: 0.01 mm

**Table 4 sensors-22-04554-t004:** Classification parameter settings for the two-layer BiLSTM neural network.

Layer	Parameter Options	Value
Input layer	Sequence length	Longest
	Batch size	27
	Learning rate	0.0001
	Input per sequence	170
	Feature vector	1 dimension
Hidden layer	BiLSTM layer	Longest
	Hidden node	(2/3) × (input size per series) [3]
	Activation function	SoftMax
Dropout layer	Dropout	0.2
Output layer	LSTM model	Many to one
	Output class	Model 1 = 26 classes Model 2 = 40 classes Model 3 = 72 classes

**Table 5 sensors-22-04554-t005:** Performance comparison of signed-letter recognition (letters A–Z).

Reference	Accuracy (%)	Error (%)	Precision (%)	Recall (%)	F1-Score (%)	SD (%)
[37]	93.81	6.19	-	-	-	-
[3]	96.07	3.93	-	-	-	-
Proposed method	97.34	2.66	97.39	97.34	97.36	0.26

**Table 6 sensors-22-04554-t006:** Performance comparison in the recognition of 40 double-hand dynamic ASL words.

Reference	Accuracy (%)	Error (%)	Precision (%)	Recall (%)	F1-Score (%)	SD (%)
[12]	97.98	2.02	96.76	97.49	96.97	-
Proposed method	98.52	1.48	98.56	98.52	98.54	0.22

**Table 7 sensors-22-04554-t007:** Performance comparison in the recognition of signed-words (72 words, including single and double-hand ASL words).

Method	Accuracy (%)	Error (%)	Precision (%)	Recall (%)	F1-Score (%)	SD (%)
Proposed method	96.99	3.01	97.01	96.99	97.00	1.01

**Table 8 sensors-22-04554-t008:** Performance comparison via an ablation test of different feature combinations in signed-letter recognition (letters A–Z).

Feature Extraction	Accuracy (%)	Precision (%)	Recall (%)	F1-Score (%)	SD (%)
$H_{2}$ $\;+\;H_{3}$	92.38	92.67	92.38	92.52	0.51
$H_{2}$ $\;+\;H_{3}$ $\;+\;H_{4}$	96.41	96.48	96.41	96.44	0.29
$H_{1}$ $\;+\;H_{2}$ $\;+\;H_{3}$ $\;+\;H_{4}$	97.34	97.39	97.34	97.36	0.26

**Table 9 sensors-22-04554-t009:** Performance comparison of ablation test of different feature combinations in 40 double-hand dynamic ASL words.

Feature Extraction	Accuracy (%)	Precision (%)	Recall (%)	F1-Score (%)	SD (%)
$H_{2}$ $\;+\;H_{3}$	86.95	88.95	86.95	87.94	0.60
$H_{2}$ $\;+\;H_{3}$ $\;+\;H_{4}$	95.51	95.88	95.51	95.69	0.34
$H_{1}$ $\;+\;H_{2}$ $\;+\;H_{3}\;$ $\;+\;H_{4}$	98.52	98.56	98.52	98.54	0.22

**Table 10 sensors-22-04554-t010:** Signed-word recognition based on single-hand data based on a backhand view and using 5-fold cross validation.

Single Hand Approach									
G.	Words	Acc. (%)	Error (%)	SD (%)	G.	Words	Acc. (%)	Error (%)	SD (%)
1	Mouse	96.20	Lonely (2.2), Nephew (1.6)	1.17	10	Respect	96.40	Are (2.6), Nephew (1)	1.20
	Lonely	96.60	Mouse (2.2), Niece (1.2)	1.20		Are	97.00	Respect (1), True (1), Mouse (1)	0.89
2	Grandfather	95.60	Grandmother (4), spit (0.3), vehicle (0.1)	1.36	11	Endorse	97.20	Latin (1.8), Am (1)	0.40
	Grandmother	97.20	Grandfather (2), spit (0.8)	0.75		Latin	96.20	Endorse (2.8), Nephew (1)	1.83
3	Tend	96.20	Delicious (3.2), Vehicle (0.4), Spit (0.2)	1.94	12	Shave	97.40	Yesterday (1.6), Fruit (0.6), Earing (0.4)	0.80
	Delicious	96.60	Tend (2.8), Spit (0.4), Vehicle (0.2)	1.36		Yesterday	96.60	Shave (2.8), Onion (0.6)	0.80
4	Hear	97.60	Whisper (1.8), Earring (0.3), Hair (0.3)	0.49	13	Apple	96.40	Onion (2.4), Niece (0.6), Eagle (0.6)	1.20
	Whisper	96.20	Hear (2.4), Fox (0.6), Grandmother (0.4), Fruit (0.4)	2.04		Onion	97.60	Apple (2.2), Yesterday (0.2)	0.49
5	Better	97.00	Forget (2.2), Saturdays (0.8)	1.09	14	Deny	97.40	Drop (2.2), Spit (0.4)	0.80
	Forget	96.20	Better (2.8), Nephew (1)	2.64		Drop	98.00	Deny (1.2), Vehicle (0.8)	0.89
6	Fox	96.60	Fruit (2.4), Earring (0.4), Hair (0.3), Whisper (0.3)	0.80	15	Niece	96.00	Nephew (2.4), Lonely (1.6)	2.45
	Fruit	97.40	Fox (1.6), Whisper (1)	0.49		Nephew	97.60	Niece (1.2), Mouse (1), Respect (0.2)	1.20
7	Past	96.80	Know (2.2), Onion (0.6), Apple (0.4)	0.98	16	Eagle	96.80	Egypt (2), Onion (0.9), Apple (0.3)	0.98
	Know	98.00	Past (2)	0.63		Egypt	97.20	Eagle (1.6), Hair (0.7), Latin (0.5)	0.40
8	Spit	95.60	Grandmother (3.4), Vehicle (1)	1.62	17	Am	98.00	True (1), Latin (1)	0.63
	Vehicle	95.80	Spit (4.2)	1.60		True	98.20	Am (1.2), Spit (0.6)	0.75
9	Earring	96.20	Hair (2), Fox (1.4), Hear (0.4)	0.97	18	Saturdays	97.00	South (2.4), Lonely (0.6)	1.26
	Hair	96.80	Earring (1.8), Fox (1), Hear (0.4)	0.75		South	96.60	Saturdays (2.8), Am (0.2), Lonely (0.2), True (0.2)	1.36

**Table 11 sensors-22-04554-t011:** Signed-word recognition based on double-hand data based on a backhand view and using 5-fold cross validation.

Double Hands Approach									
G.	Words	Acc. (%)	Error (%)	SD (%)	G.	Words	Acc. (%)	Error (%)	SD (%)
1	Holiday	98.20	Retired (1), Fire (0.5), Embarrass (0.3)	0.74	10	Until	96.80	To (2.2), Keep (0.6), Pure (0.4)	1.47
	Retired	97.00	Holiday (2), Fire (1)	1.09		To	97.60	Until (1.4), Pure (0.5), Keep (0.5)	1.36
2	Bath	96.40	Drum (3.2), Act (0.4)	0.49	11	Embarrass	97.80	Fire (1.2), Holiday (1)	0.98
	Drum	96.40	Bath (2.6), Act (0.6), Science (0.4)	0.80		Fire	95.60	Embarrass (2.4), Introduce (1.2), Convince (0.8)	0.80
3	Act	96.60	Science (2.4), Drum (0.7), Bath (0.3)	1.49	12	Cool	96.20	Fear (2.8), Shock (0.7), Sweat (0.3)	1.33
	Science	98.60	Act (1.4).	0.80		Fear	97.00	Cool (2), Street (1)	1.09
4	Carve	98.40	Page (1), Bath (0.6)	0.49	13	Father	96.60	Mother (2.6), Check (0.4), Fire (0.4)	0.80
	Page	97.00	Carve (2), Bath (1)	0.63		Mother	97.20	Father (1.8), Check (0.7), Fire (0.3)	0.40
5	Major	97.00	Street (2), Convince (1)	1.67	14	Check	97.20	Pay (2.2), Convince (0.6)	1.17
	Street	95.80	Major (2.6), Convince (1), Fear (0.6)	1.72		Pay	97.60	Check (1.4), Clean (0.5), Laid off (0.5)	1.20
6	Convince	96.20	Introduce (2.4), Major (0.5), Pay (0.5), Fire (0.4)	1.47	15	Apply	97.80	Plug (2), Keep (0.1), Pure (0.1)	0.98
	Introduce	96.80	Convince (3), Fire (0.2)	0.75		Plug	98.20	Apply (1.4), Keep (0.4)	0.40
7	Clean	97.00	Laid off (2.4), Pay (0.6)	0.74	16	Shock	98.00	Sweat (1), Fear (0.5), Cool (0.5)	0.63
	Laid off	98.20	Clean (1.2), Pay (0.4), Page (0.2)	0.40		Sweat	97.00	Shock (1.8), Cool (0.7), Fear (0.5)	0.33
8	Brother	98.60	Sister (1), Check (0.2), Keep (0.2)	0.49	17	Society	96.20	Team (2.8), Drum (1)	0.75
	Sister	98.00	Brother (1.4), Keep (0.4), Pure (0.2)	0.63		Team	97.20	Society (1.8), Science (1)	0.98
9	Awake	96.20	Surprise (2.8), Embarrass (1)	0.40	18	Keep	96.80	Sister (2), Pure (1), Brother (0.2)	1.17
	Surprise	96.80	Awake (2.2), Embarrass (1)	0.40		Pure	97.20	Keep (2.4), Sister (0.2), Brother (0.2)	1.17

## Data Availability

Not applicable.

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
