# Peer review of "Backhand-Approach-Based American Sign Language Words Recognition Using Spatial-Temporal Body Parts and Hand Relationship Patterns"

_sensors, 2022, doi:10.3390/s22124554_

Round 1

Reviewer 1 Report

The authors propose a very interesting work on gestures recognition for sign language.

I really like works that pushes forward our ability to solve more practical problems. 

My main suggestions are in the following:

Abstract

Despite having different hand positions, sign words based on a backhand approach suf- fer from misclassification due to having similar shape, rotation, and hand movement. Therefore, the system gives less accurate results.

The authors should improve the abstract. For example, the first two sentences are unclear. To which system they refer? In which context we are? Only after some sentences it is clearer to the reader.

Introduction

The introduction should contain: (a) background (b) motivations and objectives (c) proposed approach (d) gaps filled in the literature (e) major contributions/novelty of the paper (f) structure of the paper. The authors should emphasize all these points in the introduction. Furthermore, all the related works should be placed in a separated section.

Related works

The works mentioned are proper for this kind of work. But, although this will make the paper longer, I kindly suggest the authors to provide more background regarding gestures recognition in general, a field that attracted the attention of many scientists for potential helpful applications. In this way the paper will become more substantial. The gesture recognition has been applied in different contexts for different tasks , see for example: https://doi.org/10.1016/j.neucom.2019.07.103, https://doi.org/10.1145/3191755, https://doi.org/10.1145/3266302.3266308, 10.1109/FG.2019.8756513, https://doi.org/10.1007/978-981-15-5580-0_15, https://doi.org/10.1007/s11042-020-10446-y, 10.1109/ACCESS.2020.3032140. Only after this introductory part, then the focus can go to gestures recognition via leap motion analysis for sign language. I also suggest to add such seminal and very highly cited work https://doi.org/10.1007/978-94-015-8935-2_10

In the abstract the authors state  

[...] The method is further developed on 26 American sign language (ASL) letters and 40 double-hand dynamic ASL words data sets, which improve the conventional method by 1.27% and 0.54%, respectively. [...]

Have they measured the significance of this difference? Is statistically significant? Authors should also perform some kind of ablation test to understand what kind of features have led to the improvement and provide an interpretation of the results.

Experiments 

Authors must add more criteria for evaluating their proposals. The sole accuracy is not enough. Confusion matrix, F-score, Precision, Recall, Error etc should be added and compared with literature's.

Conclusion

Authors must include a dedicated paragraph on the expected impact of their solution, as well as limitations of the present work and future steps of their research.

Author Response

Dear Editor

    I appreciated your kind cooperation for reviewing and giving results of major revision for our paper entitled “Backhand-Approach-Based American Sign Language Words Recognition Using Spatial-Temporal Body Parts and Hand Relationship Patterns.”. We tried to revise the manuscript according to the reviewers and editor comments as follows.

Kosin Chamnongthai (Corresponding author)

Reviewer: #1:  The authors propose a very interesting work on gestures recognition for sign language. I really like works that pushes forward our ability to solve more practical problems.  My main suggestions are in the following:

Answer: We appreciate the reviewers for their constructive comments and suggestions.

Response:  We has undergone English language editing by MDPI.

Comments:

1.1) (Abstract) “Despite having different hand positions, sign words based on a backhand approach suffer from misclassification due to having similar shape, rotation, and hand movement. Therefore, the system gives less accurate results.” The authors should improve the abstract. For example, the first two sentences are unclear. To which system they refer? In which context we are? Only after some sentences it is clearer to the reader.  

Answer: Thank you so much for your suggestion. We have restructured the sentence for better understanding.

Response:   Page 1 in abstract part, Line 1- 15 from top.

1.2) (Introduction) The introduction should contain: (a) background (b) motivations and objectives (c) proposed approach (d) gaps filled in the literature (e) major contributions/novelty of the paper (f) structure of the paper. The authors should emphasize all these points in the introduction. Furthermore, all the related works should be placed in a separated section.

Answer: We agree that the previous version was not clear for the introduction part. Therefore, we have revised this section according to the comments already.

Response:  1.) Page 1 in Introduction part  

  • background [ Page 1, Line 15- 18 from top]
  • motivations and objectives [ Page 1 of  Line 4 from Introduction section  – Page 2 of Line 3 from top ]
  • proposed approach [ Page 2, Line 19-28 from top ]
  • gaps filled in the literature [ Page 2, Line 4-19 from top ]
  • major contributions/novelty of the paper [ Page 2, Line 29-36 from top ]

                    2.) Page 2, in Related works

1.3) (Related works) The works mentioned are proper for this kind of work. But, although this will make the paper longer, I kindly suggest the authors to provide more background regarding gestures recognition in general, a field that attracted the attention of many scientists for potential helpful applications. In this way the paper will become more substantial. The gesture recognition has been applied in different contexts for different tasks , see for example: https://doi.org/10.1016/j.neucom.2019.07.103, https://doi.org/10.1145/3191755, https://doi.org/10.1145/3266302.3266308, 10.1109/FG.2019.8756513, https://doi.org/10.1007/978-981-15-5580-0_15, https://doi.org/10.1007/s11042-020-10446-y, 10.1109/ACCESS.2020.3032140. Only after this introductory part, then the focus can go to gestures recognition via leap motion analysis for sign language. I also suggest to add such seminal and very highly cited work https://doi.org/10.1007/978-94-015-8935-2_10.

Answer:  We agree with the reviewer for adding more background regarding gesture recognition and the gesture recognition applied in different contexts for different tasks.

Response:

  1. Background [ Page 1, Line 1- 4 from Introduction section ]
  2. Different contexts for different tasks [ Page 2, 4-14 from top]
  3. https://doi.org/10.1145/3191755 : Page 2, Line 4-8 from top, reference of [7].
  4. https://doi.org/10.1016/j.neucom.2019.07.103: Page 2, Line 10-11 from top, reference of [8].
  5. https://doi.org/10.1145/3266302.3266308: Page 2, Line 11 from top, reference of [9].
  6. https://doi.org/10.1109/FG.2019.8756513: Page 2, Line 12 from top, reference of [10].
  7. https://doi.org/10.1007/978-94-015-8935-2_10: Page 2, Line 12 from bottom, reference of [14].
  8. https://doi.org/10.1007/s11042-020-10446-y: Page 2, Line 8-10 from bottom, reference of [20].
  9. https://doi.org/10.1109/ACCESS.2020.3032140: Page 3, Line 6-8 from top, reference of [34].
  10. https://doi.org/10.1007/978-981-15-5580-0_15: Page 3, Line 6-9 from top, reference of [35].

1.4) ( Abstract ) [...] The method is further developed on 26 American sign language (ASL) letters and 40 double-hand dynamic ASL words data sets, which improve the conventional method by 1.27% and 0.54%, respectively. [...]Have they measured the significance of this difference? Is statistically significant? Authors should also perform some kind of ablation test to understand what kind of features have led to the improvement and provide an interpretation of the results.

Answer: Thanks for your kind reminders. We revised the sentence as follows:

Response:  1. Page 1 in abstract, Line 9-15 from top.

  1. Page 15, Line 1-31 from bottom.
  2. Page 15, Table 5.
  3. Page 16, Table 6, Table 7, and Table 8.
  4. Page 16, Line 1- 36 from top.

1.5) (Experiments) Authors must add more criteria for evaluating their proposals. The sole accuracy is not enough. Confusion matrix, F-score, Precision, Recall, Error etc should be added and compared with literature's.

Answer: We agree with the reviewer for adding the Confusion matrix, F-score, Precision, Recall, and Error to improve a manuscript.

Response: 1.  Page 15, Table 5.

  1. Page 16, Table 6, Table 7, and Table 8.
  2. Confusion matrix: 1.) Page18, Figure 14.

1.6) (Conclusion) Authors must include a dedicated paragraph on the expected impact of their solution, as well as limitations of the present work and future steps of their research.

Answer: Thank you so much for your comments. Sorry for not describing it before. Now we have added the sentence showing the expected impact, limitations, and future works as follows.

Response: 1. Page 20, Line 1-22  from top.

Reviewer 2 Report

The paper aims to classify ASL word using LSTM. The paper is well organized. Few minor English improvements can be performed. I have few comments

·         Why LSTM was selected? Why the shown structure of LSTM was chosen?

·         Few terminologies mentioned in the figures are not defined well for instance drop out layer in figure 12 is not explained. What is the value and how it was selected.? Please add the actual sequence length as well.

·         In my opinion, informed consent of the involved human participant might be required. Please check with the editor.

·         Since authors also collected the data by themselves, the actual experimental setup should also be shown.

·         Add few study related challenges at the end

Author Response

Dear Editor

    I appreciated your kind cooperation for reviewing and giving results of major revision for our paper entitled “Backhand-Approach-Based American Sign Language Words Recognition Using Spatial-Temporal Body Parts and Hand Relationship Patterns.”. We tried to revise the manuscript according to the reviewers and editor comments as follows.

Kosin Chamnongthai (Corresponding author)

Reviewer: #2:
The paper aims to classify ASL word using LSTM. The paper is well organized. Few minor English improvements can be performed. I have few comments

Answer: We appreciate the reviewers for their constructive comments and suggestions.

Response:  We has undergone English language editing by MDPI.

2.1)  Why LSTM was selected? Why the shown structure of LSTM was chosen?

Answer: Thank you very much for your valuable suggestion. We have replaced it with a new sentence and figure with a better understanding.

Response:    1. Page 11, Line 1 – 10 from bottom.

  1. Page 12, Line 1-4 from top.
  2. Page 12, Figure 12.

2.2) Few terminologies mentioned in the figures are not defined well for instance drop out layer in figure 12 is not explained. What is the value and how it was selected.? Please add the actual sequence length as well.

Answer: Thank you so much for your minute observation and valuable comments. We have mentioned it in the revised manuscript.

Response:     1.  Page 12, Line 5-26 from top.

  1. Page 13, Line 1-7 from top.
  2. Page 14, Table 4.

2.3) In my opinion, informed consent of the involved human participant might be required. Please check with the editor.

Answer: Thank you for the advice.  We have attached informed consent with the revised manuscript.

Response: Attach File name “Informed Consent.pdf” (10 teachers )

2.4) Since authors also collected the data by themselves, the actual experimental setup should also be shown.

Answer: We agree with the reviewer for adding the figure of the actual experimental setup.

Response:  1. Page 13, Line 14-16 from top.

  1.  Page 13, Figure 13.

2.5) Add few study related challenges at the end.

Answer: We very much appreciate the reviewer’s suggestion, and we have the addition of a few study-related challenges.

Response:  1. Page 20, Line 14-22 from top.

Round 2

Reviewer 1 Report

The authors have replied to all my comments. Well done! Good job!

Hope to read and review your future works.